# Abrupt weakening of deep Atlantic circulation at the last glacial inception

Yuxin Zhou [1,2,3,4] ✉, Jerry F. McManus [1,2], Celeste T. Pallone [1,2,5], Timothy C. Kenna[1], Gabriel A. Weinstein[1,2] & Herman Garcia[1,2,6]

Deglaciations and glacial inceptions are the two equally important transitional periods that bridge the glacial and interglacial climate states, yet our understanding of deglaciations far exceeds that of glacial inceptions. Substantial variations in deep ocean circulation accompanied the last deglaciation, and model simulations recently suggested that a weakening of the Atlantic Meridional Overturning Circulation (AMOC) also occurred at the last glacial inception (LGI; 113-119 thousand years ago), yet evidence of such a change remains inconclusive. Here, we report three Pa/Th records from the western and central North Atlantic that display an abrupt weakening of the AMOC at the LGI. The magnitude of the reconstructed AMOC weakening approaches but never reaches the level of disruptions associated with the Heinrich ice discharge events. Our results may highlight a unique period of orbitally forced abrupt circulation changes and the importance of ocean processes in setting atmospheric $CO_2$ changes in motion.

During the last 800 thousand years (kyr), the Earth's climate oscillated between 100-kyr glaciations and relatively brief interglacial intervals of warmth[1]. If not for anthropogenic influence, the next climate event after the current interglacial would be the transition to a glacial period[2]. Any changes observed during the past glacial inception offer potential insights into how the transitions between two very distinct climate states, interglacial and glacial periods, took place. Before the LGI, the cryosphere configuration was analogous to that of today, while the sea level was several meters higher than today[3,4] and the global mean temperature was likely 1–1.5 °C higher than pre-industrial levels[5]. After the LGI, ice sheets started to regrow in the Northern Hemisphere, and would eventually lower the sea level by -120 m[6].

Our understanding of glacial inceptions is far inferior to that of deglaciations (Supplementary Fig. 1), and many gaps remain. Recently, a seminal Earth system model study indicated that the AMOC abruptly weakened during the LGI[7]. However, past studies of the production of North Atlantic Deep Water (NADW) during the LGI have not achieved a consensus, with some indicating active overturning[8,9] and others

suggesting diminished NADW[10–13]. Evidence of polar and subpolar cooling during the LGI, of which the AMOC weakening is a sufficient but not necessary cause, is observed in a Greenland ice core record[14] and in some reconstructed North Atlantic sea surface temperature records[13,15] but not others[16–18].

In the North Atlantic, the ratio of radioisotopes $^{231}Pa$ and $^{230}Th$ in bulk sediment is a dynamic tracer sensitive to the AMOC strength changes[19–22]. Unlike their radioactive decay parents, $^{238}U$ and $^{235}U$, which are homogeneously distributed in seawater, $^{231}Pa$ and $^{230}Th$ are readily scavenged by particles raining down through the water column. Because of a difference in their timescale of removal, a strong AMOC preferentially exports $^{231}Pa$ out of the North Atlantic[23,24] and leaves behind a low $^{231}Pa/^{230}Th$ (hereafter Pa/Th) in underlying sediments[19,20,22,23]. In turn, a weakened AMOC leads to a high Pa/Th approaching the production ratio (0.093) in sediments deposited in the deep North Atlantic.

Here, we report Pa/Th in three sediment cores collected from sites in the western and central North Atlantic (Fig. 1). IODP Site U1313 (41°

[1]Lamont-Doherty Earth Observatory of Columbia University, Palisades, NY, USA. [2]Dept. of Earth and Environmental Sciences, Columbia University, New York, NY, USA. [3]Present address: School of Earth and Atmospheric Sciences, Georgia Institute of Technology, Atlanta, GA, USA. [4]Present address: Department of Physical Oceanography, Woods Hole Oceanographic Institution, Woods Hole, MA, USA. [5]Present address: Department of Earth, Atmospheric and Planetary Sciences, Massachusetts Institute of Technology, Cambridge, MA, USA. [6]Present address: Department of Geological Sciences, University of Colorado Boulder, Boulder, CO, USA. ✉e-mail: xiaoxin.yo@gmail.com

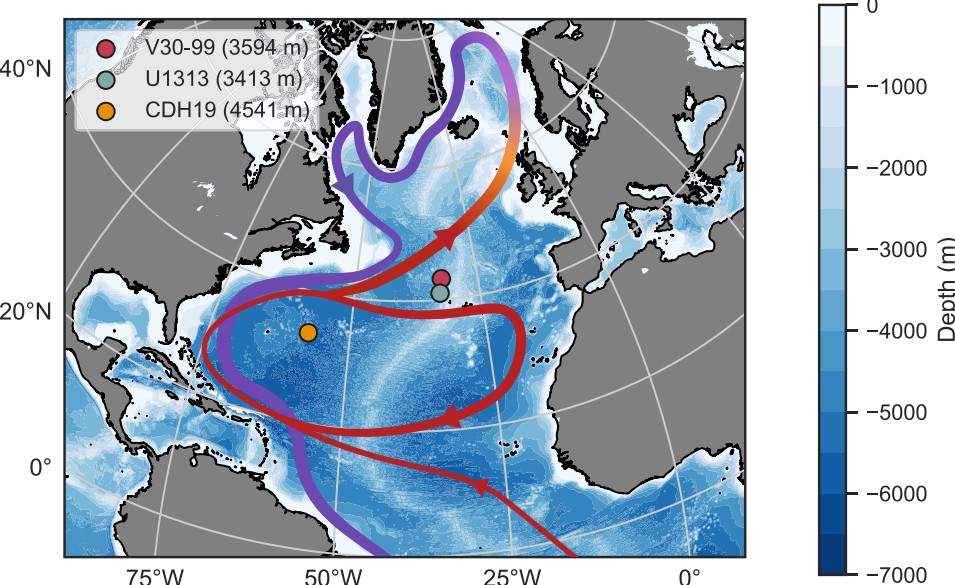

**Fig. 1 | Map of the cores used in this study.** The red and purple ribbons are the simplified warm surface currents and cold bottom flows, respectively, with the arrows marking the direction of the flows.

0.081′ N, 32° 57.421′ W, water depth 3413 m) and V30-099 (43°08.9′ N, 32°26.9′ W, water depth 3594 m) sit on the western flank of the Mid-Atlantic Ridge. Core KNR191-CDH19 was retrieved from the Bermuda Rise (33° 41.443′ N; 57° 34.559′ W, water depth 4541 m). These cores were chosen because they are from locations shown to record circulation dynamics and water mass mixing[19,22,25].

## Results

The three benthic $\delta^{18}O$ records show distinctive features in common that facilitated the target alignment to establish a shared temporal framework (Fig. 2B). The variability of the benthic $\delta^{18}O$ record from V30-99 suggests the sediment from this core may have experienced post-depositional mixing, possibly as a result of the core's relatively low sedimentation rate (~1.5 cm/kyr) and hence susceptibility to bioturbation (Fig. 2B). However, the V30-99 benthic $\delta^{18}O$–based age model is supported by an independent age model based on the constant deposition of $^{230}Th$ excess and a fixed focusing factor during the study interval[26] (Supplementary Fig. 2). The age models created by the two methods differ by at most 847 years. Additionally, we generated the benthic $\delta^{18}O$ record for the entire last glacial cycle in V30-99, further bolstering the identification of Marine Isotope Stages (MIS) 5 d and 5e in this core (Supplementary Fig. 3).

The benthic $\delta^{18}O$-based age models allow us to compare the three Pa/Th records on a common age scale (Fig. 2C–E). A consistent pattern emerges, indicating that Pa/Th increased at the LGI at all three sites, rising close to, but never quite reaching, the production ratio of 0.093. The elevated Pa/Th contrasts with the relatively low Pa/Th at each location during MIS 5e and after the LGI. In V30-99, Pa/Th increased to the production ratio at ~135 thousand years ago (ka), probably a signal of Heinrich event 11 during the penultimate deglaciation[13]. During the span of MIS 5e, all three Pa/Th records show a decreasing trend starting from an already low Pa/Th, likely indicating a continued AMOC strengthening after the recovery from Heinrich event 11. The 2σ uncertainty of Pa/Th is relatively high in CDH19 because the sediments have a lower percentage of scavenged $^{230}Th$ relative to the total $^{230}Th$ measured. The scatter plots between the preserved opal content and the Pa/Th data show a weakly negative relationship in V30-99 and a weakly positive relationship (R2 = 0.03019 and R2 = 0.1892, respectively) in U1313 and CDH19 (Supplementary Fig. 4). A comparison between the Pa/Th and

preserved opal time series in V30-99 shows that the opal content stays around 1.5% during 100-135 ka despite Pa/Th increases during the LGI and H11 (~135 ka) (Supplementary Fig. 5). In U1313, the opal content shows relatively more variability but generally is around 2%, although the opal content measurements missed the Pa/Th increases (Supplementary Fig. 5). In CDH19, the opal content is 2–6 % and does not seem to covary with Pa/Th (Supplementary Fig. 5). Notably, the increase in opal content after 110 ka is not associated with a concurrent increase in Pa/Th. Our comparison of the opal content and Pa/Th data thus indicates that the opal contents in the three cores are generally low, and the small variations in opal content are unlikely to explain the observed changes in Pa/Th, indicating that the observed Pa/Th increases primarily reflect changes in circulation rather than the preferential scavenging of $^{231}Pa$ by biogenic opal[27].

In addition to opal flux, the basin-wide total particle flux is indicated as another factor to potentially bias the Pa/Th proxy[28]. We calculated the $^{230}Th$-normalized particle flux at our three sites (see "Methods"). During the LGI, the particle flux increased at CDH19, was in an increasing trend but did not reach its maximum at V30-99, and did not increase at U1313 (Supplementary Fig. 6). The scatter plots between particle flux and Pa/Th (Supplementary Fig. 7) show a weak relationship between particle flux and Pa/Th in every case. At CDH19, the only core where the relationship is positive, the $R^2$ value is 0.07. Since these are correlations, they do not require any causality, but they provide a maximum estimate of the influence of one variable on the other. That means that particle flux has at most a 7% influence on the variance of sedimentary Pa/Th in those cases, and possibly less, from the perspective of linear modeling with least squares estimation. We infer that 93% or more of the influence on Pa/Th derives from something other than particle flux, which we interpret to be changing ocean circulation. Although we can't absolutely rule out the possibility that there are other influences, the sedimentary data are inconsistent with a dominant influence of particle flux.

The timing of the Pa/Th increase is roughly in the middle of the transition from MIS 5e to 5 d. In U1313 and CDH19, the Pa/Th increase occurs after the respective increasing trends in benthic $\delta^{18}O$ from MIS 5e to 5 d are underway. In V30-99, the Pa/Th increase is nominally spread over a relatively long time (~8 kyr, compared to the <1-kyr duration in U1313 and CDH18), again a likely sign of the potential influence of post-depositional sediment mixing.

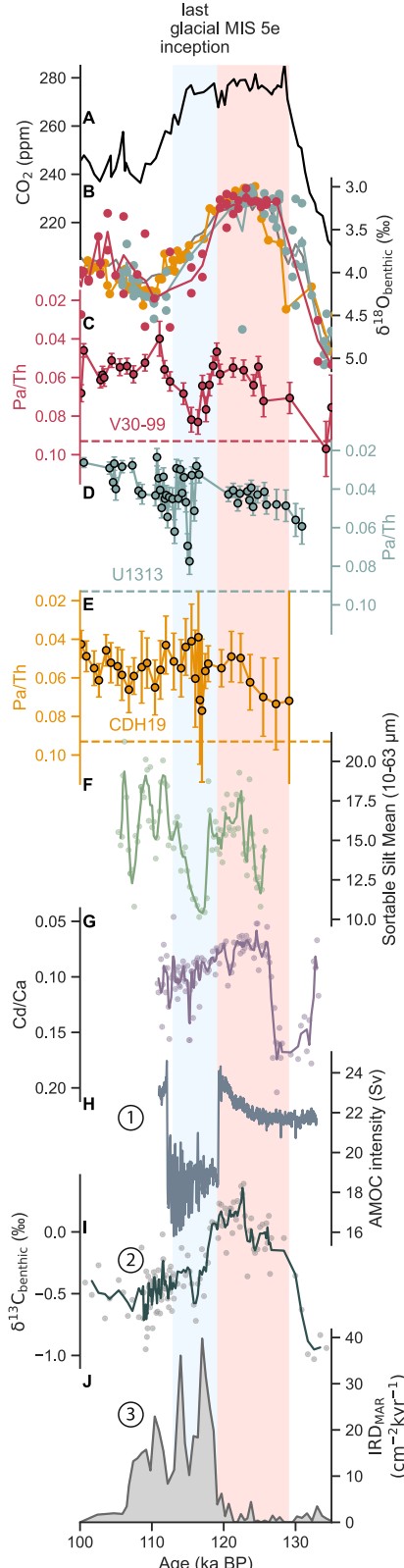

**Fig. 2 | Benthic foraminifera δ18O and Pa/Th results compared to other last glacial inception abrupt changes. A** Atmospheric CO₂[55]. **B** Benthic foraminifera δ18O results from this study. The colored dots are individual measurements. The colored lines average multiple data points at the same depth if they are available. The gray line is LR04[1]. **C–E** Pa/Th results with the 2σ error bars. Notice the y-axes are upside down. The horizontal dashed lines are the Pa/Th production ratio (0.093). **B–E** are all from this study, except CDH19 benthic δ18O in (**B**) is from ref. 21 (**F**) Sortable silt data (dots) and three-point average (line) from ref. 12 (**G**) Cd/Ca data (dots) and three-point average (line) from ref. 10 (**H**) The Atlantic Meridional Overturning Circulation intensity, defined as the maximum overturning stream-function in the North Atlantic[7]. **I** The benthic δ13C record from MD02-2448 in the South Indian Ocean[11]. The line is the three-point moving average of the raw data (dots). **J** Southern Ocean ice-rafted debris mass accumulation rate record from APcomp[39]. In (**H–J**), the numbers mark the corresponding Proposals 1–3 (see text). The blue shading is the last glacial inception (113–119 ka). The pink shading is the last interglacial (119–129 ka).

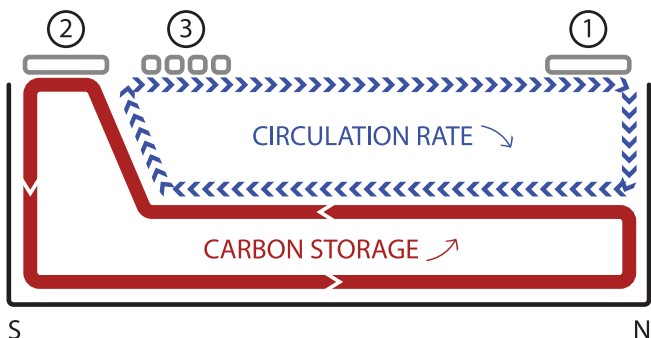

**Fig. 3 | Schematic north-south transect of the abrupt Atlantic Meridional Overturning Circulation (AMOC) slowdown during the last glacial inception (LGI) and its potential causes and impact.** The numbers mark the corresponding Proposals 1 (North Atlantic sea ice), 2 (Southern Ocean sea ice), and 3 (Antarctic iceberg) of AMOC slowdown during the LGI. As a result, the upper overturning cell weakens, and the lower cell sequesters a greater amount of dissolved inorganic carbon due to increased respired carbon accumulation and/or northward Antarctic Bottom Water expansion, leading to the inception of atmospheric CO₂ drawdown after the last interglacial.

extreme than during Heinrich events. Nevertheless, the broad geographic range covered by our core sites suggests that the LGI circulation disruption was likely a basin-wide event. Possibly because of the relatively small amplitude of Pa/Th changes compared to Heinrich events as well as the abruptness of the event, previous Pa/Th reconstructions did not observe this episode[20,29,30] (Supplementary Fig. 8). The AMOC weakening observed in our Pa/Th records fits the observed slowdown in deep current speed[12] and an increase in the proportion of southern sourced waters[10] (Fig. 2F, G). We used the accompanying benthic δ18O of these two studies to update the age models of the two records for consistency. The benthic δ18O records have been aligned to LR04 using BIGMACS via the same procedure as the three cores of this study (Supplementary Figs. 9 and 10).

What could have caused the AMOC disruption during the LGI? Here we summarize three leading proposals, one based on modeling and the other two drawn from sedimentary observations (Fig. 3). Proposal 1 focuses on orbital forcing's influence on the North Atlantic sea ice[7] (Fig. 2H). LOVECLIM1.3, an Earth system model of intermediate complexity, was used to demonstrate that the AMOC abruptly declines after an interglacial when northern hemisphere summer insolation dips below a threshold. As insolation decreases, sea ice starts to expand in the subpolar North Atlantic where deep convection takes place. The sea ice cover insulates the surface ocean against heat loss to the atmosphere. The resulting warming and increased buoyancy

## Discussion

Because Pa/Th can be used as a proxy of the overall AMOC strength, the Pa/Th increases at the LGI at all three study sites indicate an extensive AMOC weakening at the time. Since the LGI Pa/Th increases never reach the production ratio, as have been observed within the Heinrich layers[19–21], the circulation disruption was probably less

suppress deep convection[31] and lead to an abrupt weakening of the AMOC until insolation bounces back above the threshold (Fig. 2H).

Proposal 2 involves sea-ice formation in the Southern Ocean[11,32]. The downward trending insolation at the LGI induces an equatorward shift of the westerlies[33], misaligning the westerlies with the Antarctic Circumpolar Current, reducing the Ekman transport, and suppressing the upwelling of relatively warm deep waters[34]. Less upwelling of warm water, together with cooling in the southern hemisphere high latitudes[35], allows sea ice to expand. The resulting brine injection intensifies the Antarctic Bottom Water (AABW) production (observation: ref. 11; Fig. 2I; modeling: ref. 36). As the density contrast between the deep and abyssal overturning cells increases, the expansion of a denser AABW forces the AMOC lower limb to shoal and suppresses its overturning[32,37,38].

Lastly, an increase in Antarctic icebergs reaching and melting at the Agulhas region was observed during the LGI (Fig. 2J). Proposal 3 points to Antarctic iceberg melting as a mechanism that would advect positive buoyancy flux into the upper limb of the AMOC and therefore cause its disruption[39]. In a sense, the mechanism of Proposal 3 is the opposite of the "Agulhas leakage" that injects warm and saline water into the Atlantic[40]. The icebergs reaching the Agulhas region increase the freshwater input to the Atlantic. An increase in sea-ice extent and cooling in the Southern Ocean, attributed to decreased insolation[33] (Proposal 2), are suggested to improve the survival of Antarctic icebergs and facilitate a northward shift in iceberg trajectories[39].

Our study cannot definitively determine which of the three proposals is the most likely scenario, but we find tentative clues in the particle flux supporting Proposal 1. Specifically, [230]Th-normalized particle flux measurements from site CDH19 show an increase during the LGI concurrent with the Pa/Th rise. This particle flux increase could indicate increased export productivity, ice rafting of detrital materials, dust flux, or underwater density flow. These local or regional changes are more likely to be caused by processes that originated in the Northern Hemisphere, which exists in Proposal 1 only. On the other hand, increases in particle flux are much more muted at sites U1313 and V30-99 during the LGI, and the particle flux increases also exist during other periods with little changes in Pa/Th. Therefore, the interpretation that our findings support Proposal 1 is only speculative, and additional future research on the bipolar and subpolar regions during the LGI is required to shed light on the causes of the AMOC weakening.

A common thread among the three proposals is the involvement of orbital forcing in influencing sea-ice formation and, in turn, causing AMOC disruptions. North Atlantic iceberg or meltwater discharge is not implicated in this episode of abrupt AMOC weakening. Indeed, Northern Hemisphere ice sheets were only starting to expand from their nucleation centers[41,42], and little evidence exists for a substantial North Atlantic iceberg or meltwater discharge event at the time[8,13,43,44]. Other instances of abrupt AMOC declines during Heinrich events[19,21] and the last interglacial[45,46] do not have obvious connections with orbital forcing, highlighting the LGI as a possibly unique instance of orbitally forced abrupt circulation changes.

Since the observed AMOC weakening occurred after benthic $\delta^{18}$O already started increasing from its last interglacial minimum, our results do not indicate that the AMOC slowdown initiated the LGI. Instead, orbital forcing changes alone appears to have been sufficient to initiate the LGI, as has been suggested in modeling studies[7,47]. Nevertheless, it is possible that the AMOC weakening accelerated the glacial regrowth by curtailing the northward heat transport and cooling North America for the nascent Laurentide Ice Sheet[48,49].

Our data present new benchmarks, but not direct challenges, to the "moisture initiators" hypothesis for explaining early ice sheet growth. The "moisture initiators" mechanism states that the supply of moisture towards high-latitude continents is essential for ice-sheet accretion[16–18]. The moisture could then induce snowfall and initiate glaciation during the LGI[50,51]. Because the upper limb of the AMOC transports warm surface water northward, a vigorous AMOC is argued to supply moisture towards the nucleation sites for the Laurentide Ice Sheet. Other studies emphasize the atmospheric route of moisture transport, which could have been enhanced due to an increased equator-to-pole surface temperature gradient[52–54]. Our observations suggest that the AMOC, the oceanic route of moisture transport, remained strong initially and then weakened within the period of rapid glacial expansion, if benthic $\delta^{18}$O is used as a tracer of ice volume. An increase in subpolar North Atlantic sea-ice formation, as laid out in Proposal 1, could have further suppressed the oceanic moisture supply. Yet, the circulation slowdown did not seem to interrupt the ice volume growth. Our results thus favor the atmospheric route over the oceanic one of moisture transport as a viable explanation for enhancing ice sheet growth under declining insolation.

During the LGI, atmospheric $CO_2$ remained persistently elevated for about four thousand years, even after Antarctic temperature cooled[5,55] and global ice volume began to increase[1], potentially as a result of declining obliquity[56]. The first significant drawdown of atmospheric $CO_2$ did not occur until the episode of AMOC weakening at 115 ka (Fig. 2A). This might not have been a coincidence. We propose that increased accumulation of respired carbon and/or northward AABW expansion, linked to AMOC weakening as shown by our results, could lead to $CO_2$ drawdown[32,37,38] (Fig. 3). The increased dissolved inorganic carbon (DIC) concentration of AABW would further activate the ocean alkalinity feedback via the lysocline shoaling to amplify $CO_2$ sequestration in the deep ocean[57]. This process, together with the expansion of Antarctic sea ice that could act as a lid to limit the outgassing of carbon from upwelled deep waters in the Southern Ocean[58], could explain the delayed timing of the atmospheric $CO_2$ decrease at the LGI. The LGI may thus exemplify the importance of ocean processes in setting atmospheric $CO_2$ changes in motion.

## Methods

The sediment samples from V30-99 and U1313 were freeze-dried, soaked in deionized water, and disaggregated on the Cambridge washing wheel disaggregator for an hour. The wet samples were washed through 63 μm sieves with the help of the Lamont automated sample sieving bench, and the coarse fraction retained in the sieves was dried and transferred to glass vials. The dried >63 μm fraction was again dry sieved at >150 μm and examined under a microscope. In core V30-99, the benthic foraminifera *Cibicidoides wuellerstorfi* tests were picked. In core U1313, the tests of both *Cibicidoides wuellerstorfi* and *Uvigerina spp.* were picked. The $\delta^{18}$O measurements on the benthic foraminifera tests were conducted with a Thermo Delta V Plus gas-source isotope-ratio mass spectrometer equipped with a Kiel IV individual acid-bath sample preparation device at the Lamont-Doherty Earth Observatory of Columbia University stable isotope laboratory. The $\delta^{18}$O records were corrected to *Uvigerina* using an offset of 0.64‰ for *Cibicidoides*[59]. The long-term standard deviation of $\delta^{18}$O measurements made on carbonate standard NBS19 is 0.06 ‰. At depths with abundant benthic foraminifera tests, up to four separate stable isotope analyses were carried out. In core CDH19, a benthic $\delta^{18}$O record measured on *Cibicidoides wuellerstorfi* and *Nuttallides umbonifera* was previously made public[21].

The chronostratigraphies of the three cores were established by aligning the benthic $\delta^{18}$O records to the LR04 global stack[1] using the open-source BIGMACS software (https://github.com/eilion/BIGMACS)[60]. While the automated BIGMACS alignment procedure performs generally well, because of how short the alignment period is, we also added two alignment data points using the built-in additional age control function. First, a 172 cm sediment depth in V30-99 is given an age of 118 ka. Second, 3322 cm sediment depth in CDH19 is given an age of 115 ka.

Bulk sediment samples of ~100 mg were spiked with [229]Th, [236]U, and [233]Pa, digested, purified[61], and analyzed for uranium, thorium, and

protactinium isotopic activities. Isotopes were measured on an Element 2 inductively coupled plasma mass spectrometer (ICP-MS) using either Nickle Jet sample cone and X skimmer cone (for Pa) or Standard sample cone and X skimmer cone (for U and Th), and coupled to either a CETAC™ Aridus desolvating nebulizer (For Pa) or an ESI-PC3 Peltier cooled cyclonic spray chamber (for U and Th) at the Lamont-Doherty Earth Observatory of Columbia University. $^{238}$U and $^{232}$Th were measured in analog mode, and the rest of the isotopes were made in ion counting mode. Tail corrections and mass bias corrections were made, and the analog/counting gain was calculated[61]. Every batch of 18 samples was accompanied by a procedural blank, an internal standard called the North Atlantic Internal Mega Standard (NAIMS), and a $^{233}$Pa/$^{231}$Pa mixture solution to track the decay of the $^{233}$Pa spike since its creation. The procedural blanks from the 11 batches contribute, on average, 3% of the $^{238}$U measured from samples, 0.4% of $^{230}$Th and $^{232}$Th, and 0.9% of $^{231}$Pa. The repeated measurements of internal standard NAIMS from the 11 batches determined the 1σ precision to be 8.5% for $^{238}$U, 3.4% for $^{230}$Th, 8.2% for $^{232}$Th, and 9.6% for $^{231}$Pa.

The radioactive isotopes $^{235}$U (half-life: 704 million years) and $^{238}$U (half-life: 4.5 billion years) are highly soluble in seawater and have long residence times (~ 400 kyr). In contrast, their decay products, $^{231}$Pa (half-life: 32.7 kyr) and $^{230}$Th (half-life: 75.584 kyr), are highly insoluble and readily scavenged by sinking particles[62]. The residence time of $^{231}$Pa (100–200 years) is shorter than that of $^{230}$Th (20–40 years)[63] and approaches the Atlantic deep water transit time. As a result, a vigorous AMOC, such as the condition today, exports approximately half of the $^{231}$Pa produced in the Atlantic basin towards the Southern Ocean[23,24]. A weaker AMOC leads to more $^{231}$Pa being deposited in the Atlantic sediments, pushing Pa/Th higher to approach its production ratio (0.093).

The measured bulk sediment concentrations of $^{230}$Th and $^{231}$Pa include contributions of the detrital (produced from the radioactive decay of U in mineral lattices) and authigenic (from the radioactive decay of U that precipitated from the soluble form U(VI) to its insoluble form U(IV) in anoxic, reducing sediments) fractions. In calculating the scavenged portions of $^{230}$Th and $^{231}$Pa, these other two sources need to be accounted for and corrected. Detrital $^{230}$Th and $^{231}$Pa can be estimated from the measured concentration of $^{232}$Th, which is entirely of detrital origin[64]. We apply site-specific lithogenic $^{238}$U/$^{232}$Th activity ratios, using $^{234}$U/$^{238}$U to gauge the presence of authigenic U[65]. After excluding samples with $^{234}$U/$^{238}$U more than 0.96 to account for the loss of 4% of $^{234}$U from the detrital sediments by alpha-recoil[65], we also excluded one $^{238}$U/$^{232}$Th outlier data point in V30-99 (encircled in Supplementary Fig. 11B). We additionally found samples with abnormally low $^{234}$U/$^{238}$U in U1313 compared to the adjacent samples, all from a single batch of measurements, which we have excluded as well (encircled in Supplementary Fig. 11C). The average of the remaining $^{238}$U/$^{232}$Th data points in each core is used as the local detrital $^{238}$U/$^{232}$Th. The resulting local detrital $^{238}$U/$^{232}$Th estimates are 0.48 at V30-99, 0.57 at U1313, and 0.52 at CDH19. We additionally apply a lithogenic $^{230}$Th/$^{238}$U activity ratio of 0.81[66] and a natural $^{235}$U/$^{238}$U activity ratio of 0.046[67]. We note that Pa/Th is not as affected by the specific choice of lithogenic $^{238}$U/$^{232}$Th activity ratio as other uranium series proxies[68,69]. Authigenic $^{230}$Th and $^{231}$Pa are estimated from the non-detrital portion of $^{238}$U and $^{235}$U by assuming a seawater $^{234}$U/$^{238}$U activity ratio of 1.1468[70] and correcting for time passed since uranium precipitation.

The 1σ uncertainty of the measured isotopes is estimated with the standard deviation of the 200 scans of the isotopes by the ICP-MS. We detect and remove outliers of the 200 scans using the modified z-score[71]: $M_i = \frac{x_i - \tilde{x}}{MAD}$, where $M_i$ is the modified z-score, $x_i$ is the value to be analyzed, $\tilde{x}$ is the median, and MAD is the median absolute deviation. Outliers are defined as values with a modified z-score greater than 2. The uncertainty propagation considers the ICP-MS intensity drift

during the 200 scans of each sample and standard (National Institute of Standards and Technology Standard Reference Materials Uranium Standard or SRM). The accepted ratio of the SRM $^{238}$U/$^{235}$U is 137.7145, and we apply a relative standard deviation of 0.5% to the ratio in error propagation. We estimate the relative 1σ uncertainty of the lithogenic $^{238}$U/$^{232}$Th and $^{230}$Th/$^{238}$U activity ratios to be 5%, also propagated while calculating the uncertainty. The conversion from raw protactinium counting data to activities and associated error propagation has been packaged into a Python script named PaxsPy, accessible at https://github.com/yz3062/PaxsPy. The counterpart Python script for thorium has previously been published[66] at https://github.com/yz3062/ThxsPy.

To test whether the preferential scavenging of $^{231}$Pa by biogenic opal influenced our Pa/Th results[27], we measured biogenic opal in V30-99, U1313, and CDH19 following two established methods, one utilizing spectrometry[72] and the other with inductively coupled plasma optical emission spectroscopy (ICP-OES)[73]. At the Bermuda Rise site of CDH19, Pa/Th has previously been shown to be little affected by biogenic opal[20]. For the spectrometry method, bulk sediment samples of ~100 mg were mixed with sodium carbonate and heated at 85 °C for 5 h to extract opal. Silica concentration was measured with a molybdate-blue spectrophotometry method. For the ICP-OES method, a sub-sample of 0.3 ml from the leachate was added to 10 ml of Milli-Q water in 15 mL centrifuge tubes and neutralized with HNO$_3$ to pH = 6. The final volume was then adjusted to 12 mL with additional Milli-Q water. Silicon concentrations were quantified using an Agilent 720 (axial) ICP-OES housed on the LDEO campus of Columbia University. The standard curve used was brought up in a matching Na$_2$CO$_3$ matrix to minimize any matrix effects between the samples and standards. The standard curve encompassed the expected range of the samples. A drift solution, made of a mixture of samples, was run after every 5th sample to monitor and correct for any changes in sensitivity. Silicon was measured on four different wavelengths (185.005 nm, 250.690 nm, 251.611 nm, and 288.158 nm), all of which yielded good signal intensity. The concentration of Silica was calculated based on each individual wavelength, and they were then averaged together for a final value. Percent opal was estimated from Si concentrations using a formula weight conversion factor of 2.4.

To assess whether particle flux could affect our Pa/Th results, we calculated particle flux F = β * Z / $^{230}$Th$_{xs,0}$, where F is the vertical particle flux, β is the $^{230}$Th production rate in the water column, Z is the water depth, and $^{230}$Th$_{xs,0}$ is the excess $^{230}$Th corrected for radioactive decay (i.e., the denominator of Pa/Th).

## Data availability

The uranium series data, benthic δ$^{18}$O, and opal content generated in this study have been deposited in a Figshare repository[74] (https://doi.org/10.6084/m9.figshare.25330645.v3).

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

## Acknowledgments

We thank Martin Fleisher, Annette Lott, Andie Munoz, Alyssa Ramirez, Chandler Morris, Alissa Lampert, Bennett Slibeck, and Christopher Bauco for assistance in the laboratory. We also thank Qiuzhen Yin for sharing the glacial inception modeling data. This research used samples and/or data provided by the International Ocean Discovery Program (IODP). This research was funded in part by NSF grant AGS 16-35019 and OCE 24-42513 to J.F.M. and by the IODP Schlanger Fellowship to Y.Z.

## Author contributions

Y.Z. and J.F.M. jointly initiated the research project. C.T.P., T.C.K., G.A.W., and H.G. contributed to the data collection. Y.Z., J.F.M., C.T.P., T.C.K., G.A.W., and H.G. contributed to the interpretation of the results and writing of the manuscript.

## Competing interests

The authors declare no competing interests.
