## [Transparent Peer Review file · Nature Communications]

Abrupt weakening of deep Atlantic circulation at the last glacial inception

Corresponding Author: Dr Yuxin Zhou

Version 0:

Reviewer comments:

Reviewer #1

(Remarks to the Author)
Review report

Zhou et al. reconstructed the strength of AMOC during Last Glacial Inception (LGI) from Pa/Th records in 3 sediment cores retrieved from the North Atlantic. They interpreted the Pa/Th ratios being close to, but not reaching, their production ratio of 0.093 during LGI as weakening of AMOC, but not to the extent of the weakening observed elsewhere during Heinrich events. Based on these interpretations, the authors discussed three existing proposals for the causes of the AMOC disruption during the LGI, and implications to the importance of ocean processes in setting atmospheric CO₂ changes. The authors carried out reconstruction of AMOC for a period (LGI) that is important but understudied compared with other periods. Given that the results are fully justified, this work could fill in the knowledge gap about AMOC and its implication to the climate relevant to the current and future climate changes. The study produced relevant datasets to support the discussion, with good amount of Pa/Th data, supported by benthic d¹⁸O and biogenic opal. The manuscript was well presented with clear and accessible texts and figures. Previous studies were well considered, and cited appropriately to support the discussion of this study. Before accepting the manuscript for publication, however, there are a few major points that need to be addressed.

1. Interpretation of Pa/Th records as variation in AMOC intensity: The results and discussion rely strongly on AMOC as the primary control of the variations in the Pa/Th ratios observed in the sedimentary records. However sedimentary Pa/Th records are also influenced by particle flux and composition. The authors ruled out the influence of opal flux based on little relationship between opal flux and Pa/Th ratios (Line 82-84). There are two points that require further clarification here:

1.1. Biogenic opal fluxes were calculated from opal weight percentage normalized to ²³⁰Th concentrations. However, correction of the presence of detrital ²³⁰Th in the measured ²³⁰Th assumes the same ²³⁸U/²³²Th ratio of 0.48 for all three cores (line 360). This assumption might not be valid due to the different sedimentary settings the three cores were located, with one of the cores (CDH19) in a different basin from the other two cores. The choice of this ratio as the authors pointed out might not affect Pa/Th ratios significantly, but might affect ²³⁰Th concentrations alone, and hence calculation of opal flux using ²³⁰Th concentrations. It therefore requires either justification of using the same ²³⁸U/²³²Th ratio to correct for ²³⁰Th concentrations for all three cores, or adoption of local ²³⁸U/²³²Th instead (Bourne et al., 2012).

1.2. Apart from biogenic opal flux, particle flux (total of CaCO₃, detrital, organic matter, and opal), i.e. scavenging intensity not differentiating composition, could also affect sedimentary Pa/Th ratios (e.g. Missiaen et al., 2020). There lacks discussion of particle flux and justification that variation in particle flux could be ruled out for interpretation of Pa/Th in the manuscript.

2. Pa, Th, and U data: The description of the methods to obtain Pa, Th and U data in Methods does not allow evaluation of the quality of the data. I suggest that the details below be included.

2.1. Measurement on ICP-MS: there lacks information on instrument set-up, ICP-MS method adopted, corrections if any, procedural blanks, etc. If this is a routine measurement published in previous studies, such studies should be provided as reference.

2.2. Uncertainty and error propagation: details of how the uncertainties were assessed are lacking. Brief summary of uncertainties for Pa/Th data in all three cores should be provided. Although the authors provided the link to the Python package used for error propagation, brief description of the types of errors that were accounted for and the propagation methodology should be included in Methods.

3. Discussion: I have a major concern here about the writing of this section. The authors described mechanisms proposed in previous studies in great length (three long paragraphs), but there is limited discussion (relative to the length of the description of the mechanisms) about the results of this work for or against the mechanisms. Similar to this is the discussion of how the results of this study explain moisture initiators, and linkage to CO₂ drawdown. For instance, the authors pointed out the coincidence between AMOC weakening and delayed CO₂ drawdown, but continued with explaining the delay of CO₂ drawdown by mechanisms proposed in other studies. The results of this study could have been given more focus and discussion, for instance, to establish the linkage between AMOC weakening and CO₂ decrease.

Minor comments:

1. Figure 1: check orange symbol for CDH19 on the map is the same size as the other two (it seems smaller).

2. Line 46: suggest changing “radioisotopic decay” to “radioactive decay”.

3. Line 46-47: suggest changing the sentence “unlike the radioactive decay parents ²³⁸U and ²³⁵U, ... , their decay products ²³¹Pa and ²³⁰Th are readily... “.
to, “unlike their radioactive decay parents ²³⁸U and ²³⁵U, ... , ²³¹Pa and ²³⁰Th are readily.... “.

4. Figure S4: error bars should be included.

References:

Bourne, M.D. et al. (2012) ‘Improved determination of marine sedimentation rates using ²³⁰Thxs’, *Geochemistry, Geophysics, Geosystems*, 13(9). Available at: <https://doi.org/https://doi.org/10.1029/2012GC004295>.

Missiaen, L. et al. (2020) ‘Modelling the impact of biogenic particle flux intensity and composition on sedimentary Pa/Th’, *Quaternary Science Reviews*, 240, p. 106394. Available at: <https://doi.org/https://doi.org/10.1016/j.quascirev.2020.106394>.

Reviewer #2

(Remarks to the Author)

This study investigates the signature of sedimentary ²³¹Pa/²³⁰Th extracted from cores in the deep North Atlantic during the Last Glacial Inception (119-113 ka) in order to understand, how the AMOC responded to this climatic event that has, in contrast to glacial terminations, not received much attention despite its potential importance and implications for the future evolution of climate, given that the next glacial inception will occur in the not so distant future. The exact mechanisms are important to understand given that it is not clear what the consequences of man-made climate change during the upcoming inception will be.

The manuscript presents three new ²³¹Pa/²³⁰Th data sets showing positive excursions of different duration during the LGI period, which are, however, not very pronounced in at least two of these three records. In many cases the data presented during the LGI are not significantly different outside analytical uncertainties from the data before or afterwards. This is particularly the case for Core CDH19, in which the data during MIS5e within error were as high as during the LGI. In the ²³¹Pa/²³⁰Th data set presented for CDH19 and Site U1313, the LGI excursion is defined by only two data points within the LGI. What duration of a phase of weakened AMOC would these data indicate (are we really talking about 1 kyr)? In this respect, it is therefore important to present any supporting evidence available for simultaneous climatic events accompanying a weaker AMOC during the LGI from other data set as part of the manuscript, which the authors have tried to do in Fig. 2. However, the $\delta^{13}\text{C}$ data and IRD data sets presented show a high variability and to me it is unclear if there are any significant minima that could have occurred contemporaneously with a significant short term AMOC weakening during the LGI. The timing of the changes of the measured proxies and the duration of potentially LGI-related changes differs strongly and it is therefore not clear if they reflect the same events and forcings.

In addition to the abovementioned limitations, there is obviously no evidence for a major weakening of the AMOC during LGI in any other North Atlantic Pa/Th data sets (see figure S5). Why is this or, if turned around, why is the LGI signal visible in only two cores of this study (the CDH19 Pa/Th increase is not statistically significant)? The North Atlantic basin-wide integrated Pa/Th data should show a Pa/Th increase during the LGI or a part of it to support AMOC weakening, which I cannot see. In addition, the modelled significant AMOC reduction (Yin et al., 2021) comprises the entire LGI period, which is definitely not indicated by the Pa/Th data. And in some records, such as MD95-2037, the MIS5e Pa/Th data are even higher than those during the LGI. In that respect, instead of only showing the correlations in S4, I would have liked to see a plot, in which the opal time series of the cores are compared to the Pa/Th time series in order to check if there were no simultaneous changes in both data sets that may explain at least a part of the observed Pa/Th variability.

In summary, this is overall a well written manuscript but, as outlined above, I am unfortunately not convinced of the significance of the variations in the new and old Pa/Th time series as indicators of a pronounced AMOC change during the LGI. I therefore must recommend rejection of the manuscript. In order to further pursue the investigations of glacial inceptions and their impact on AMOC changes in the frame of future work, which I clearly deem worthwhile, it would be worth looking at other and better resolved sediment cores and/or other glacial inception periods such as the transitions from MIS 5a to MIS 4

(the error bars of the Pa/Th data would be smaller because there is still more initially adsorbed ^{231}Pa present in the sediments of that age) and combining Pa/Th data with other proxies for AMOC changes such as Nd isotopes.

Additional smaller comments:

Line 156:...could have been enhanced...

Line 159-160: I think the word "believe" should not be used here.

Version 1:

Reviewer comments:

Reviewer #1

(Remarks to the Author)

As a reviewer, I appreciate all the effort Zhou et al. have made to improve the manuscript based on the reviewer's comments. However, there is one major concern based on the new figure/data presented in the revised version (Authors' Response #2). The authors presented in Figure S6 particle fluxes for three cores (minor comment here: x axis label and label of the colour band for LGI and MIS5e were missing). After seeing this new figure, I am not convinced that the authors could rule out the influence of the particle flux in their interpretation of the Pa/Th. For core CDH19, the authors stated that high particle flux was observed and might contribute to high Pa/Th. However the authors then argued that they could be ruled out by referring to other work. McManus et al. (2014) did measure the same core as used in this study. They however presented results without high enough time resolution to provide support for this study. That is why this study providing Pa/Th records that were not presented before could fill the knowledge gap in the first place. On top of that, McManus et al. ruled out the influence of particle flux on Pa/Th by showing higher Pa/Th observed coincide with when lower particle flux was observed. The new result/figure shown in this manuscript could not provide evidence that the similar is true for when high Pa/Th was observed during LGI, with evidence however showing the opposite, i.e. high Pa/Th coinciding with high particle flux. For the same reason, Bohm et al. (2015), quoted in the manuscript, does not also support the authors' argument. Another quoted article, Henry et al. (2016) presented records spanning 20-60 kyr BP, which is beyond the period of the core in this study. I could not see how these records could be used to rule out the influence of particle flux in this study. The basin-wide particle flux, which show little change during the LGI does not really provide strong support having shown the down core records of variation in particle flux in Fig.S6. For core V30-99, there is increasing trend that coincide with the increase in Pa/Th. There lacks the discussion for this core in the revised manuscript. In summary, having seen the particle flux data shown in Figure S6. and not been convinced by the authors' approach to rule out the influence of the particle flux on interpreting Pa/Th as solely indicating AMOC strength, I could not recommend this manuscript for publication.

Minor:

Authors' response #1: The authors adopted the local detrital $^{238}\text{U}/^{232}\text{Th}$ as suggested by the reviewer to correct for the presence of detrital ^{230}Th . Their choice of local detrital $^{238}\text{U}/^{232}\text{Th}$ is based on the minimum of $^{238}\text{U}/^{232}\text{Th}$ in the core. This needs to be clarified with more details. Bourne et al. (2012) suggested that the $^{238}\text{U}/^{232}\text{Th}$ ratios that represent the local detrital $^{238}\text{U}/^{232}\text{Th}$ are those that are at the depth without the presence of authigenic U. Not having the information about the presence/absence of authigenic U in the cores, it is not possible to make sense why authors' choice of using minimum $^{238}\text{U}/^{232}\text{Th}$ in the core to represent local detrital $^{238}\text{U}/^{232}\text{Th}$.

Reviewer #2

(Remarks to the Author)

This is the revised version of a study that I reviewed previously and that investigates the signature of sedimentary $^{231}\text{Pa}/^{230}\text{Th}$ extracted from three cores in the deep North Atlantic during the Last Glacial Inception (119-113 ka) in order to understand, how the AMOC responded to this climatic event that has, in contrast to glacial terminations, not received much attention despite that the next glacial inception will occur in the not so distant future.

The authors have added some new data for comparison to their data sets and despite that the number of data points capturing the LGI Pa/Th maximum is small, the possibility is high that there indeed was an AMOC instability during the LGI. I am still not 100% convinced that the data obtained from three cores undoubtedly reflect a short AMOC weakening of up to 1800 years but at the same time I cannot disprove it. I am therefore willing to consider the possibility that it occurred. Given the importance of such a finding I think that the community should now decide itself if this weakening may have occurred based on the presented data and at best produce further well resolved $^{231}\text{Pa}/^{230}\text{Th}$ records for this period of time to support or disprove the conclusions presented in this manuscript. In the end the $^{231}\text{Pa}/^{230}\text{Th}$ proxy is not supposed to show the same signal in each and every core but it is rather the basin-integrated ^{231}Pa deficit (low $^{231}\text{Pa}/^{230}\text{Th}$) that documents a strong AMOC whereas a lack or a decrease of the basin-integrated deficit will document an AMOC weakening (see original publication on the use of $^{231}\text{Pa}/^{230}\text{Th}$ as a paleo circulation tracer in the Atlantic Ocean by Yu et al. 1996). As such, further data will improve or disprove the reliability of the presented conclusions.

In summary, based on the above considerations I am now supporting publication of the manuscript.

Minor suggestion for a change of the text in Line 161: This particle flux increase could....

Version 2:

Reviewer comments:

Reviewer #1

(Remarks to the Author)

After the previous review of the revised manuscript aimed at reconstructing the strength of AMO during Last Glacial Maximum from sedimentary Pa/Th records, the authors provided additional assessment of the particle type and rewording to address the influence of the particle flux on the interpretation of Pa/Th records. The authors also conducted reanalysis of the correction to obtain the scavenged fraction of the Pa/Th sediments, as suggested in the previous review, and updated the results in this revised version of the manuscript. Both of the work very well addressed the points raised in the previous review. I am happy to recommend the publication of this manuscript.

Minor:

Figure S6, and Figure S7: units for particle fluxes are incorrect, missing subscript (Fig. S6), and having typos (Fig. S7).

Response to comments from reviewers

Key

- Reviewers' comments
- Authors' response

Reviewer #1 (Remarks to the Author):

Review report

Zhou et al. reconstructed the strength of AMOC during Last Glacial Inception (LGI) from Pa/Th records in 3 sediment cores retrieved from the North Atlantic. They interpreted the Pa/Th ratios being close to, but not reaching, their production ratio of 0.093 during LGI as weakening of AMOC, but not to the extent of the weakening observed elsewhere during Heinrich events. Based on these interpretations, the authors discussed three existing proposals for the causes of the AMOC disruption during the LGI, and implications to the importance of ocean processes in setting atmospheric CO₂ changes. The authors carried out reconstruction of AMOC for a period (LGI) that is important but understudied compared with other periods. Given that the results are fully justified, this work could fill in the knowledge gap about AMOC and its implication to the climate relevant to the current and future climate changes. The study produced relevant datasets to support the discussion, with good amount of Pa/Th data, supported by benthic δ¹⁸O and biogenic opal. The manuscript was well presented with clear and accessible texts and figures. Previous studies were well considered, and cited appropriately to support the discussion of this study. Before accepting the manuscript for publication, however, there are a few major points that need to be addressed.

1. Interpretation of Pa/Th records as variation in AMOC intensity: The results and discussion rely strongly on AMOC as the primary control of the variations in the Pa/Th ratios observed in the sedimentary records. However sedimentary Pa/Th records are also influenced by particle flux and composition. The authors ruled out the influence of opal flux based on little relationship between opal flux and Pa/Th ratios (Line 82-84). There are two points that require further clarification here:

1.1. Biogenic opal fluxes were calculated from opal weight percentage normalized to ²³⁰Th concentrations. However, correction of the presence of detrital ²³⁰Th in the measured ²³⁰Th assumes the same ²³⁸U/²³²Th ratio of 0.48 for all three cores (line 360). This assumption might not be valid due to the different sedimentary settings the three cores were located, with one of the cores (CDH19) in a different basin from the other two cores. The choice of this ratio as the authors pointed out might not affect Pa/Th ratios significantly, but might affect ²³⁰Th concentrations alone, and hence calculation of opal flux using ²³⁰Th concentrations. It therefore requires either justification of using the same ²³⁸U/²³²Th ratio to correct for ²³⁰Th concentrations for all three cores, or adoption of local ²³⁸U/²³²Th instead (Bourne et al., 2012).

Response #1 – We agree with the reviewer that adopting local detrital ²³⁸U/²³²Th ratios makes more sense, and we have done so and updated the relevant figures and Methods (Line 438) accordingly. The detrital ²³⁸U/²³²Th, which is derived from the minimal ²³⁸U/²³²Th at each site, is

0.22 at V30-99, 0.36 at U1313, and 0.42 at CDH19. As noted by the reviewer, the resulting Pa/Th records do not deviate much from the original (see figure below for a comparison of Figure 2 using a constant 0.48 ratio for all cores (left) and using local detrital $^{238}\text{U}/^{232}\text{Th}$ ratios (right)).

The reviewer also noted that the local detrital $^{238}\text{U}/^{232}\text{Th}$ ratio changes could potentially change the opal flux via changes to the ^{230}Th -normalized mass flux. We show that the opal flux vs. Pa/Th scatter plot (Fig. S4) mostly retains the same negative trend after the detrital $^{238}\text{U}/^{232}\text{Th}$ ratio changes. In the figure below, the left panels were calculated with a constant 0.48 ratio for all cores, and the right panels were calculated using local detrital $^{238}\text{U}/^{232}\text{Th}$ ratios. Note that we made more opal measurements in response to Reviewer 2's comments, shown in the right panels.

1.2. Apart from biogenic opal flux, particle flux (total of CaCO_3 , detrital, organic matter, and opal), i.e. scavenging intensity not differentiating composition, could also affect sedimentary Pa/Th ratios (e.g. Missiaen et al., 2020). There lacks discussion of particle flux and justification that variation in particle flux could be ruled out for interpretation of Pa/Th in the manuscript.

Response #2 – We thank the reviewers for reminding us to think more deeply and comprehensively about the impact particle flux may have on Pa/Th. We have dedicated a paragraph to discuss the potential complication from particle flux (Line 98) and added a citation to Missiaen et al. (2020):

“In addition to opal flux, the basin-wide total particle flux is indicated as another factor to potentially bias the Pa/Th proxy (Missiaen et al., 2020). We calculated the ^{230}Th -normalized particle flux at our three sites (see Methods). During the LGI, the particle flux increased at CDH19, was in an increasing trend but did not reach its maximum at V30-99, and did not increase at U1313 (Fig. S6). Although the particle flux changes at CDH19 possibly has an impact on the Pa/Th record, previous Pa/Th records at Bermuda Rise (Böhm et al., 2015; Henry et al., 2016; McManus et al., 2004), where CDH19 was retrieved, have been corroborated by other proxies (Lynch-Stieglitz et al., 2014; Thiagarajan et al., 2014). A previous basin-wide compilation of particle flux (Zhou and McManus, 2024) also shows no dramatic changes during the LGI (Fig. S7).”

2. Pa, Th, and U data: The description of the methods to obtain Pa, Th and U data in Methods does not allow evaluation of the quality of the data. I suggest that the details below be included.

2.1. Measurement on ICP-MS: there lacks information on instrument set-up, ICP-MS method adopted, corrections if any, procedural blanks, etc. If this is a routine measurement published in previous studies, such studies should be provided as reference.

Response #3 – Agreed. We have added information on the ICP-MS procedure (Line 388):

“Isotopes were measured on an Element 2 inductively coupled plasma mass spectrometer (ICP-MS) using either Nickle Jet sample cone and X skimmer cone (for Pa) or Standard sample cone and X skimmer cone (for U and Th), and coupled to either a CETAC™ Aridus desolvating nebulizer (For Pa) or an ESI-PC3 Peltier cooled cyclonic spray chamber (for U and Th) at the Lamont-Doherty Earth Observatory of Columbia University. ^{238}U and ^{232}Th were measured in analog mode, and the rest of the isotopes were made in ion counting mode. Tail corrections and mass bias corrections were made, and the analog/counting gain was calculated (Fleisher and Anderson, 2003). Every batch of 18 samples was accompanied by a procedural blank, an internal standard called the North Atlantic Internal Mega Standard (NAIMS), and a $^{233}\text{Pa}/^{231}\text{Pa}$ mixture solution to track the decay of the ^{233}Pa spike since its creation. The procedural blanks from the 11 batches contribute, on average, 3% of the ^{238}U measured from samples, 0.4% of ^{230}Th and ^{232}Th , and 0.9% of ^{231}Pa . The repeated measurements of internal standard NAIMS from the 11 batches determined the 1σ precision to be 8.5% for ^{238}U , 3.4% for ^{230}Th , 8.2% for ^{232}Th , and 9.6% for ^{231}Pa .”

2.2. Uncertainty and error propagation: details of how the uncertainties were assessed are lacking. Brief summary of uncertainties for Pa/Th data in all three cores should be provided. Although the authors provided the link to the Python package used for error propagation, brief description of the types of errors that were accounted for and the propagation methodology should be included in Methods.

Response #4 – We thank the reviewer for reminding us to be transparent with the error propagation procedure and have added text discussing how the uncertainty is constrained by this study (L425):

“The 1σ uncertainty of the measured isotopes is estimated with the standard deviation of the 200 scans of the isotopes by the ICP-MS. We detect and remove outliers of the 200 scans using the modified z-score (Iglewicz and Hoaglin, 1993): $M_i = \frac{x_i - \tilde{x}}{MAD}$, where M_i is the modified z-score, x_i is the value to be analyzed, \tilde{x} is the median, and MAD is the median absolute deviation. Outliers are defined as values with a modified z-score greater than 2. The uncertainty propagation considers the ICP-MS intensity drift during the 200 scans of each sample and standard (National Institute of Standards and Technology Standard Reference Materials Uranium Standard or SRM). The accepted ratio of the SRM $^{238}\text{U}/^{235}\text{U}$ is 137.7145, and we apply a relative standard deviation of 0.5% to the ratio in error propagation. We estimate the relative 1σ uncertainty of the lithogenic $^{238}\text{U}/^{232}\text{Th}$ and $^{230}\text{Th}/^{238}\text{U}$ activity ratios to be 5%, also propagated while calculating the uncertainty. The conversion from raw protactinium counting data to activities and associated error propagation has been packaged into a Python script named PaxisPy, accessible at

<https://github.com/yz3062/PaxsPy>. The counterpart Python script for thorium has previously been published (Zhou et al., 2021) at <https://github.com/yz3062/ThxsPy>.”

3. Discussion: I have a major concern here about the writing of this section. The authors described mechanisms proposed in previous studies in great length (three long paragraphs), but there is limited discussion (relative to the length of the description of the mechanisms) about the results of this work for or against the mechanisms. Similar to this is the discussion of how the results of this study explain moisture initiators, and linkage to CO₂ drawdown. For instance, the authors pointed out the coincidence between AMOC weakening and delayed CO₂ drawdown, but continued with explaining the delay of CO₂ drawdown by mechanisms proposed in other studies. The results of this study could have been given more focus and discussion, for instance, to establish the linkage between AMOC weakening and CO₂ decrease.

Response #5 – Agreed. We have added a paragraph (L158) describing how our results could potentially support Proposal 1:

“Our study cannot definitively determine which of the three proposals is the most likely scenario, but we find tentative clues in the particle flux supporting Proposal 1. Specifically, ²³⁰Th-normalized particle flux measurements from site CDH19 show an increase during the LGI concurrent with the Pa/Th rise. Particle flux increase could indicate increased export productivity, ice rafting of detrital materials, dust flux, or underwater density flow. These local or regional changes are more likely to be caused by processes that originated in the Northern Hemisphere, which exists in Proposal 1 only. On the other hand, increases in particle flux are much more muted at sites U1313 and V30-99 during the LGI, and the particle flux increases also exist during other periods with little changes in Pa/Th. Therefore, the interpretation that our findings support Proposal 1 is only speculative, and future research on the bi-polar and -subpolar regions during the LGI will shed light on the causes of the AMOC weakening.”

We have also rephrased the text linking AMOC weakening and CO₂ decrease to put the results of this study at the center of the discussion (L205).

Minor comments:

Figure 1: check orange symbol for CDH19 on the map is the same size as the other two (it seems smaller).

Response #6 – We checked, and at least programmatically, the symbols are supposed to be the same sizes. We do agree with the reviewer that the orange symbol looks smaller and we are puzzled by this discrepancy. It is possible that the darker background makes the orange circle seem smaller.

Line 46: suggest changing “radioisotopic decay” to “radioactive decay”.

Response #7 – Done.

Line 46-47: suggest changing the sentence “unlike the radioactive decay parents ²³⁸U and ²³⁵U, ... , their decay products ²³¹Pa and ²³⁰Th are readily... “.

to, “unlike their radioactive decay parents ^{238}U and ^{235}U , ... , ^{231}Pa and ^{230}Th are readily....
“.

Response #8 – Done.

Figure S4: error bars should be included.

Response #9 – Done.

References:

Bourne, M.D. et al. (2012) ‘Improved determination of marine sedimentation rates using ^{230}Th ’, *Geochemistry, Geophysics, Geosystems*, 13(9). Available at: <https://doi.org/https://doi.org/10.1029/2012GC004295>.

Missiaen, L. et al. (2020) ‘Modelling the impact of biogenic particle flux intensity and composition on sedimentary Pa/Th’, *Quaternary Science Reviews*, 240, p. 106394. Available at: <https://doi.org/https://doi.org/10.1016/j.quascirev.2020.106394>.

Reviewer #2 (Remarks to the Author):

This study investigates the signature of sedimentary $^{231}\text{Pa}/^{230}\text{Th}$ extracted from cores in the deep North Atlantic during the Last Glacial Inception (119-113 ka) in order to understand, how the AMOC responded to this climatic event that has, in contrast to glacial terminations, not received much attention despite its potential importance and implications for the future evolution of climate, given that the next glacial inception will occur in the not so distant future. The exact mechanisms are important to understand given that it is not clear what the consequences of man-made climate change during the upcoming inception will be.

The manuscript presents three new $^{231}\text{Pa}/^{230}\text{Th}$ data sets showing positive excursions of different duration during the LGI period, which are, however, not very pronounced in at least two of these three records. In many cases the data presented during the LGI are not significantly different outside analytical uncertainties from the data before or afterwards. This is particularly the case for Core CDH19, in which the data during MIS5e within error were as high as during the LGI. In the $^{231}\text{Pa}/^{230}\text{Th}$ data set presented for CDH19 and Site U1313, the LGI excursion is defined by only two data points within the LGI. What duration of a phase of weakened AMOC would these data indicate (are we really talking about 1 kyr)? In this respect, it is therefore important to present any supporting evidence available for simultaneous climatic events accompanying a weaker AMOC during the LGI from other data set as part of the manuscript, which the authors have tried to do in Fig. 2. However, the $\delta^{13}\text{C}$ data and IRD data sets presented show a high variability and to me it is unclear if there are any significant minima that could have occurred contemporaneously with a significant short term AMOC weakening during the LGI. The timing of the changes of the measured proxies and the duration of potentially LGI-related changes differs strongly and it is therefore not clear if they reflect the same events and forcings.

Response #10 – We thank the reviewers for reminding us to interrogate the veracity of the data, especially given the observed changes are relatively small and short-lived. To answer the reviewer's question about the duration of the observed AMOC changes, in V30-99, the two Pa/Th data points during the LGI represent 1780 years. In U1313, they represent 543 years. In CDH19, they represent 623 years. The short duration of the AMOC weakening is similarly captured by one or two Pa/Th data points in previous studies, such as during H3-10 in Böhm et al. (2015) and non-HE stadials in Henry et al. (2016), demonstrating the sensitivity of Pa/Th as a proxy to short-term, abrupt AMOC changes.

We added two North Atlantic records, a sortable silt record from Hall et al. (1998) and a Cd/Ca record from Adkins et al. (1997), to our Fig. 2 (see below). We used the accompanying benthic $\delta^{18}\text{O}$ of these two studies to update the age models of the two records for consistency. The benthic $\delta^{18}\text{O}$ records have been aligned to LR04 using BIGMACS via the same procedure as the three cores of this study (Fig S9, 10).

Sortable silt is a proxy for bottom flow speed, while Cd/Ca is conventionally used as a water mass tracer because of its empirical covariation with the deepwater nutrient level. Because these proxies target different paleoceanographic processes than Pa/Th, the circulation changes implied by these two records differ from our Pa/Th data in timing and magnitude. Nevertheless, they provide more corroborative evidence of a period of circulation instability during the LGI.

The reviewer rightly pointed out that the duration of AMOC weakening simulated by Yin et al. (2021) is much longer than our results suggest (~7 kyr). However, the speed of AMOC recovery can be highly dependent on the model setup. In the simulations run by Yin et al. (2021), only insolation and CO_2 were allowed to vary with time. Using the same version of the model (LOVECLIM1.3) as Yin et al. (2021) but including a realistic NH ice sheet evolution, Loutre et al. (Loutre et al., 2014) found an LGI AMOC weakening of a similar magnitude as Yi et al. (2021) but much shorter in duration (~1.5 kyr) that better resembles our Pa/Th reconstructions (their topoGR run as seen in Fig. 4b red curve within). Loutre et al. (2014) suggest that the NH ice sheets, as they grow in size during the LGI, have an increasingly stabilizing effect on the AMOC due to the lower sea surface temperature in the vicinity of the ice sheets that is conducive to deepwater formation. Therefore, the longer duration of the LGI AMOC weakening simulated by Yin et al. (2021) could be due to the experimental design. Indeed, following the Yin et al. (2021) proposal that an insolation threshold exists below which the AMOC destabilizes, the duration of the AMOC weakening could vary depending on the insolation threshold, which is controlled by the model parameters. The simulation of Yin et al. (2021) could easily reproduce the timing and magnitude of the AMOC weakening we observe. This is because a minimum in summer insolation at 65°N occurs at 114 ka, concurrent with or slightly before the Pa/Th increases seen in our records. A lower insolation threshold for AMOC weakening, perhaps when the model includes NH ice sheet evolution, would then be consistent with our observation.

The IRD flux record (Starr et al., 2021) is also longer in duration than the Pa/Th increases in our records. The longer duration of the IRD flux peak could be due to the way IRD flux was calculated. The IRD flux in that study was derived from the benthic $\delta^{18}\text{O}$ -based age model and the resultant linear sedimentation rate. This method of calculating flux leads to over-smoothing the flux because of the limited age model resolution achievable and the interpolation required –

underestimating the flux peaks and overestimating the flux troughs. This is especially true during IRD-rich zones in the sediment, where benthic foraminifera can be rare, and the resolution of the benthic $\delta^{18}\text{O}$ record within the zone becomes poor. As a result, the IRD flux record (Starr et al., 2021) could potentially be consistent with the duration of the Pa/Th peaks in our records if a better flux calculation method is used.

In summary, our observed AMOC weakening during the LGI is supported by paleo-reconstructions and models in the literature. While the literature data differ slightly in timing and variability, they offer corroborative evidence of circulation instability that our new Pa/Th data bring into sharper relief.

Figure 1. Benthic foraminifera $\delta^{18}\text{O}$ and Pa/Th results compared to other LGI abrupt changes. **(A)** Atmospheric CO_2 ⁵⁸. **(B)** Benthic foraminifera $\delta^{18}\text{O}$ results from this study. The colored dots are individual measurements. The colored lines average multiple data points at the same depth if they are available. The gray line is LR04³. **(C, D, E)** Pa/Th results with the 2σ error bars. Notice the y-axes are upside down. The horizontal dashed lines are the Pa/Th production (0.093). **(B-E)** are all from this study except CDH19 benthic $\delta^{18}\text{O}$ in **(B)** is from ref. ²¹. **(F)** Sortable silt data (dots) and three-point average (line) from ref. ¹³. **(G)** Cd/Ca data (dots) and three-point average (line) from ref. ¹¹. **(H)** The AMOC intensity, defined as the maximum overturning streamfunction in the North Atlantic². **(I)** The benthic $\delta^{13}\text{C}$ record from MD02-2448 in the South Indian Ocean¹². The line is the three-point moving average of the raw data (dots). **(J)** Southern Ocean ice-rafted debris mass accumulation rate record from AP_{comp}⁴². In **(H-J)**, the numbers mark the corresponding Proposals 1-3 (see text). The blue shading is the last glacial inception (113-119 ka). The pink shading is the last interglacial (119-129 ka).

In addition to the abovementioned limitations, there is obviously no evidence for a major weakening of the AMOC during LGI in any other North Atlantic Pa/Th data sets (see figure S5). Why is this or, if turned around, why is the LGI signal visible in only two cores of this study (the CDH19 Pa/Th increase is not statistically significant)? The North Atlantic basin-wide integrated Pa/Th data should show a Pa/Th increase during the LGI or a part of it to support AMOC weakening, which I cannot see. In addition, the modelled significant AMOC reduction (Yin et al., 2021) comprises the entire LGI period, which is definitely not indicated by the Pa/Th data. And in some records, such as MD95-2037, the MIS5e Pa/Th data are even higher than those during the LGI. In that respect, instead of only showing the correlations in S4, I would have liked to see a plot, in which the opal time series of the cores are compared to the Pa/Th time series in order to check if there were no simultaneous changes in both data sets that may explain at least a part of the observed Pa/Th variability

Response #11 – We agree with the reviewer that our Pa/Th data should be compared with the preserved opal content to check its influence. We made more opal measurements to double the amount of data points compared to our last submission. Opal measurements are now also available for core CDH19. A comparison between the Pa/Th and preserved opal time series in V30-99 shows that the opal content stays around 1.5% during 100-135 ka despite Pa/Th increases during the LGI and H11 (~135 ka). In U1313, the opal content shows relatively more variability but generally is around 2%, although the opal content measurements missed the Pa/Th increases. In CDH19, the opal content is 2-6 % and does not seem to covary with Pa/Th. Notably, the increase in opal content after 110 ka is not associated with a concurrent increase in

Figure S5. Pa/Th and opal time series comparison. (A, C, E) Pa/Th data during the LGI. (B, D, F) Opal content during the same period.

Pa/Th. We present opal content to better illustrate the relatively low preserved opal content found in our cores, as the concentration accounts for the fact that the relative opal content in the total particle flux can potentially influence Pa/Th without changes in the opal flux (Lippold et al., 2019, 2016).

The scatter plots between the preserved opal content and the Pa/Th data show a weakly negative relationship in V30-99 and a weakly positive relationship ($R^2=0.0191$ and $R^2=0.1208$, respectively) in U1313 and CDH19. Our comparison of the opal content and Pa/Th data thus indicates that the opal contents in the three cores are generally low, and the variations in the opal contents are unlikely to explain the observed Pa/Th changes.

Figure S4. Scatter plots of opal content vs. Pa/Th in the three cores of our study, V30-99 (A), U1313 (B), and CDH19 (C). The inset equations are the least square linear regression fit and the associated uncertainty.

Our finding of low correlation between the preserved opal content and Pa/Th is consistent with previous studies that looked into the same correlation in western and central North Atlantic sites (Guihou et al., 2010; Lippold et al., 2019, 2012; Ng et al., 2018), including studies that investigated the same sites as our study or sites very close to ours (Guihou et al., 2011; Henry et al., 2016; Lippold et al., 2016; McManus et al., 2004). Also consistent with previous studies, we find the preserved opal content in the western and central North Atlantic is generally very low. Although variable opal preservation could complicate the interpretation of the preserved opal content, the more corrosive Antarctic Bottom Water tends to be found in the eastern North Atlantic during periods of North Atlantic Deep Water shoaling (Zhou and McManus, 2023). Additionally, our measured opal contents do not show distinct trends between the last interglacial and the LGI. Only minimal opal dissolution probably occurred during the last interglacial when a vigorous AMOC bathed our sites with newly formed deep water. Opal dissolution would have to match the increase in opal production to produce a mostly constant opal content from the last interglacial to the LGI. Therefore, although we cannot rule out that opal dissolution occurred at the three sites of our study and impacted our interpretation of the relationship between opal content and Pa/Th, we deem the scenario unlikely.

The reviewer is right to point out that there was no indication of an AMOC slowdown in the existing literature. The lower resolution of the existing Pa/Th records might have prevented the identification of this event. Since the Pa/Th increase during the LGI does not reach the production ratio and is relatively short-lived, it is best captured by high-resolution Pa/Th records.

In summary, this is overall a well written manuscript but, as outlined above, I am unfortunately not convinced of the significance of the variations in the new and old Pa/Th time series as indicators of a pronounced AMOC change during the LGI. I therefore must recommend rejection of the manuscript. In order to further pursue the investigations of glacial inceptions and their impact on AMOC changes in the frame of future work, which I clearly deem worthwhile, it would be worth looking at other and better resolved sediment cores and/or other glacial inception periods such as the transitions from MIS 5a to MIS 4 (the error bars of the Pa/Th data would be smaller because there is still more initially adsorbed ²³¹Pa present in the sediments of that age) and combining Pa/Th data with other proxies for AMOC changes such as Nd isotopes.

Additional smaller comments:

Line 156:...could have been enhanced...

Response #12 – Done.

Line 159-160: I think the word "believe" should not be used here.

Response #13 – Agreed. We have rephrased this sentence to be:

"An increase in subpolar North Atlantic sea-ice formation, as laid out in Proposal 1, could have further suppressed the oceanic moisture supply."

Lastly, we identified a typo in the caption for Fig. 2. The colored lines in the benthic $\delta^{18}\text{O}$ panels are the averages of multiple data points at the same depth if available, not the three-point averages.

References

- Böhm, E., Lippold, J., Gutjahr, M., Frank, M., Blaser, P., Antz, B., Fohlmeister, J., Frank, N., Andersen, M.B., Deininger, M., Bohm, E., Lippold, J., Gutjahr, M., Frank, M., Blaser, P., Antz, B., Fohlmeister, J., Frank, N., Andersen, M.B., Deininger, M., 2015. Strong and deep Atlantic meridional overturning circulation during the last glacial cycle. *Nature* 517, 73–76. <https://doi.org/10.1038/nature14059>
- Fleisher, M.Q., Anderson, R.F., 2003. Assessing the collection efficiency of Ross Sea sediment traps using ²³⁰Th and ²³¹Pa. *Deep Sea Research Part II: Topical Studies in Oceanography* 50, 693–712. [https://doi.org/10.1016/S0967-0645\(02\)00591-X](https://doi.org/10.1016/S0967-0645(02)00591-X)
- Guihou, A., Pichat, S., Govin, A., Nave, S., Michel, E., Duplessy, J.C., Telouk, P., Labeyrie, L., 2011. Enhanced Atlantic Meridional Overturning Circulation supports the Last Glacial Inception. *Quaternary Science Reviews* 30, 1576–1582. <https://doi.org/10.1016/j.quascirev.2011.03.017>
- Guihou, A., Pichat, S., Nave, S., Govin, A., Labeyrie, L., Michel, E., Waelbroeck, C., 2010. Late slowdown of the Atlantic Meridional Overturning Circulation during the Last Glacial

- Inception: New constraints from sedimentary (231Pa/230Th). *Earth and Planetary Science Letters* 289, 520–529. <https://doi.org/10.1016/j.epsl.2009.11.045>
- Henry, L.G., McManus, J.F., Curry, W.B., Roberts, N.L., Piotrowski, A.M., Keigwin, L.D., 2016. North Atlantic ocean circulation and abrupt climate change during the last glaciation. *Science* 353, 470–474. <https://doi.org/10.1126/science.aaf5529>
- Iglewicz, B., Hoaglin, D.C., 1993. Volume 16: how to detect and handle outliers. Quality Press.
- Lippold, J., Gutjahr, M., Blaser, P., Christner, E., de Carvalho Ferreira, M.L., Mulitza, S., Christl, M., Wombacher, F., Böhm, E., Antz, B., Cartapanis, O., Vogel, H., Jaccard, S.L., 2016. Deep water provenance and dynamics of the (de)glacial Atlantic meridional overturning circulation. *Earth and Planetary Science Letters* 445, 68–78. <https://doi.org/10.1016/j.epsl.2016.04.013>
- Lippold, J., Luo, Y., Francois, R., Allen, S.E., Gherardi, J., Pichat, S., Hickey, B., Schulz, H., 2012. Strength and geometry of the glacial Atlantic Meridional Overturning Circulation. *Nature Geoscience* 5, 813–816. <https://doi.org/10.1038/ngeo1608>
- Lippold, J., Pöppelmeier, F., Sufke, F., Gutjahr, M., Goepfert, T.J., Blaser, P., Friedrich, O., Link, J.M., Wacker, L., Rheinberger, S., Jaccard, S.L., 2019. Constraining the Variability of the Atlantic Meridional Overturning Circulation During the Holocene. *Geophysical Research Letters* 46, 11338–11346. <https://doi.org/10.1029/2019GL084988>
- Loutre, M.F., Fichet, T., Goosse, H., Huybrechts, P., Goelzer, H., Capron, E., 2014. Factors controlling the last interglacial climate as simulated by LOVECLIM1.3. *Clim. Past* 10, 1541–1565. <https://doi.org/10.5194/cp-10-1541-2014>
- Lynch-Stieglitz, J., Schmidt, M.W., Henry, L., Curry, W.B., Skinner, L.C., Mulitza, S., Zhang, R., Chang, P., 2014. Muted change in Atlantic overturning circulation over some glacial-aged Heinrich events. *Nature Geoscience* 7, 144–150. <https://doi.org/10.1038/ngeo2045>
- McManus, J.F., Francois, R., Gherardi, J.M., Keigwin, L., Drown-Leger, S., 2004. Collapse and rapid resumption of Atlantic meridional circulation linked to deglacial climate changes. *Nature* 428, 834–837. <https://doi.org/10.1038/nature02494>
- Missiaen, L., Menviel, L.C., Meissner, K.J., Roche, D.M., Dutay, J.-C., Bouttes, N., Lhardy, F., Quiquet, A., Pichat, S., Waelbroeck, C., 2020. Modelling the impact of biogenic particle flux intensity and composition on sedimentary Pa/Th. *Quaternary Science Reviews* 240, 106394. <https://doi.org/10.1016/j.quascirev.2020.106394>
- Ng, H.C., Robinson, L.F., McManus, J.F., Mohamed, K.J., Jacobel, A.W., Ivanovic, R.F., Gregoire, L.J., Chen, T., 2018. Coherent deglacial changes in western Atlantic Ocean circulation. *Nature Communications* 9, 2947. <https://doi.org/10.1038/s41467-018-05312-3>
- Thiagarajan, N., Subhas, A.V., Southon, J.R., Eiler, J.M., Adkins, J.F., 2014. Abrupt pre-Bølling-Allerød warming and circulation changes in the deep ocean. *Nature* 511, 75–78. <https://doi.org/10.1038/nature13472>
- Zhou, Y., McManus, J.F., 2024. Heinrich event ice discharge and the fate of the Atlantic Meridional Overturning Circulation. *Science* 384, 983–986. <https://doi.org/10.1126/science.adh8369>
- Zhou, Y., McManus, J.F., 2023. Authigenic uranium deposition in the glacial North Atlantic: Implications for changes in oxygenation, carbon storage, and deep water-mass geometry. *Quaternary Science Reviews* 300, 107914. <https://doi.org/10.1016/j.quascirev.2022.107914>

Zhou, Y., McManus, J.F., Jacobel, A.W., Costa, K.M., Wang, S., Alvarez Caraveo, B., 2021.
Enhanced iceberg discharge in the western North Atlantic during all Heinrich events of
the last glaciation. *Earth and Planetary Science Letters* 564, 116910.
<https://doi.org/10.1016/j.epsl.2021.116910>

Response to comments from reviewers

Key

- Reviewers' comments
- Authors' response

Reviewer #1 (Remarks to the Author):

As a reviewer, I appreciate all the effort Zhou et al. have made to improve the manuscript based on the reviewer's comments. However, there is one major concern based on the new figure/data presented in the revised version (Authors' Response #2). The authors presented in Figure S6 particle fluxes for three cores (minor comment here: x axis label and label of the colour band for LGI and MIS5e were missing). After seeing this new figure, I am not convinced that the authors could rule out the influence of the particle flux in their interpretation of the Pa/Th. For core CDH19, the authors stated that high particle flux was observed and might contribute to high Pa/Th. However the authors then argued that they could be ruled out by referring to other work. McManus et al. (2014) did measure the same core as used in this study. They however presented results without high enough time resolution to provide support for this study. That is why this study providing Pa/Th records that were not presented before could fill the knowledge gap in the first place. On top of that, McManus et al. ruled out the influence of particle flux on Pa/Th by showing higher Pa/Th observed coincide with when lower particle flux was observed. The new result/figure shown in this manuscript could not provide evidence that the similar is true for when high Pa/Th was observed during LGI, with evidence however showing the opposite, i.e. high Pa/Th coinciding with high particle flux. For the same reason, Bohm et al. (2015), quoted in the manuscript, does not also support the authors' argument. Another quoted article, Henry et al. (2016) presented records spanning 20-60 kyr BP, which is beyond the period of the core in this study. I could not see how these records could be used to rule out the influence of particle flux in this study. The basin-wide particle flux, which show little change during the LGI does not really provide strong support having shown the down core records of variation in particle flux in Fig.S6. For core V30-99, there is increasing trend that coincide with the increase in Pa/Th. There lacks the discussion for this core in the revised manuscript. In summary, having seen the particle flux data shown in Figure S6. and not been convinced by the authors' approach to rule out the influence of the particle flux on interpreting Pa/Th as solely indicating AMOC strength, I could not recommend this manuscript for publication.

Response #1 – We agree with the reviewer that the influence of particle flux on Pa/Th deserves to be examined more closely. We changed the presentation of the manuscript to make a more effective argument (L101). We continue to believe the influence of particle flux on Pa/Th in our records is minimal.

The scatter plots between particle flux and Pa/Th (Fig. R1) show a weak relationship between particle flux and Pa/Th in every case, with the strongest correlation at site U1313, where the association is negative, the opposite of any inferred influence of particle flux.

Fig. R1. Scatter plots of particle flux and Pa/Th in the three cores of our study, V30-99 (A), U1313 (B), and CDH19 (C). The inset equations are the least square linear regression fit and the associated uncertainty. The pink symbols in each plot are the data points during the LGI interval.

At the other two sites, there is an extremely weak relationship, with R^2 values of 0.003 and 0.07. Since those are correlations, they do not require any causality at all, but they provide a maximum estimate of the influence of one variable on the other. That means that particle flux has at most a 0.3-7% influence on the variance of sedimentary Pa/Th in those cases, and possibly less, from the perspective of linear modeling with least squares estimation. We infer that 93% or more of the influence on Pa/Th derives from something other than particle flux, which we interpret to be changing ocean circulation. Although we can't absolutely rule out the possibility that there are other influences, the sedimentary data are inconsistent with a dominant influence of particle flux.

We also examined the opal, CaCO_3 , and lithogenic fluxes of V30-99 and CDH19 to discern which sedimentary type drives the particle flux changes during the LGI (Fig. R2). The CaCO_3 record from V30-99 is admittedly very low in resolution (Ruddiman and Farrell, 1996). Nevertheless, the data suggest that the causes of particle flux increases in V30-99 and CDH19 are different; CaCO_3 flux drives the particle flux increase in V30-99, while lithogenic flux drives the particle flux increase in CDH19.

Lithogenics and CaCO_3 scavenge Th more intensely than Pa (Chase and Anderson, 2004; Hayes et al., 2015; Luo and Ku, 2004; Missiaen et al., 2020a, 2020b; Rempfer et al., 2017; Scheen et al., 2025), and an increase in the fluxes of these two particle types, based on fractionation alone, should theoretically lead to a decrease in sedimentary Pa/Th. The particle types may at least partially explain the relatively weak (<7%) influence of the particle flux on sedimentary Pa/Th in V30-99 and CDH19 (Fig. R1). Previous studies have noted that the influence of particle flux on sedimentary Pa/Th can be counteracted by some particle types, including lithogenics (Siddall et al., 2005) and CaCO_3 (Missiaen et al., 2020b).

Figure R2. Comparison of particle, opal, CaCO₃, and lithogenic flux of V30-99 (left) and CDH19 (right). (A, E) Particle flux (this study). (B, F) Opal flux (this study). (C, G) CaCO₃ flux calculated from particle flux (this study) and %CaCO₃ (V30-99: Ruddiman et al., 1996; CDH19: Henry et al., 2016). (D, H) Lithogenic flux (this study) calculated from ²³²Th flux, assuming an average ²³²Th concentration of 10.7 ppm in upper continental crust (Taylor and McClelland, 1995).

Lastly, the observed LGI Pa/Th increases coincide with changes in several circulation and water mass tracers, as well as the simulation of the AMOC, which provides independent evidence indicating that the AMOC changes are the likely cause of the observed changes (Fig. 2).

Minor:

Authors' response #1: The authors adopted the local detrital ²³⁸U/²³²Th as suggested by the reviewer to correct for the presence of detrital ²³⁰Th. Their choice of local detrital ²³⁸U/²³²Th is based on the minimum of ²³⁸U/²³²Th in the core. This needs to be clarified with more details. Bourne et al. (2012) suggested that the ²³⁸U/²³²Th ratios that represent the local detrital ²³⁸U/²³²Th are those that are at the depth without the presence of authigenic U. Not having the

information about the presence/absence of authigenic U in the cores, it is not possible to make sense why authors' choice of using minimum $^{238}\text{U}/^{232}\text{Th}$ in the core to represent local detrital $^{238}\text{U}/^{232}\text{Th}$.

Response #2 – We thank the reviewer for reminding us to engage more thoroughly with the literature. Indeed, Bourne et al. (2012) suggested using $^{234}\text{U}/^{238}\text{U}$ to gauge the presence of authigenic U. Samples with lower $^{234}\text{U}/^{238}\text{U}$ are more likely to have little to no authigenic U, and the average $^{238}\text{U}/^{232}\text{Th}$ of these samples can be used to estimate the local detrital $^{238}\text{U}/^{232}\text{Th}$.

We have now adopted this approach from Bourne et al. (Fig. R3). After excluding samples with $^{234}\text{U}/^{238}\text{U}$ more than 0.96 to account for the loss of 4% of ^{234}U from the detrital sediments by alpha-recoil (Bourne et al., 2012), we also excluded one $^{238}\text{U}/^{232}\text{Th}$ outlier data point in V30-99 (encircled in Fig. R3B). We additionally found samples with abnormally low $^{234}\text{U}/^{238}\text{U}$ in U1313 compared to the adjacent samples, all from a single batch of measurements, which we have excluded as well (encircled in Fig. R3C). Despite the exclusion of more samples in U1313, the prominent pattern of $^{238}\text{U}/^{232}\text{Th}$ increase during the LGI makes the identification of authigenic U relatively straightforward.

The resulting local detrital $^{238}\text{U}/^{232}\text{Th}$ estimates are 0.48 at V30-99, 0.57 at U1313, and 0.52 at CDH19. This is in contrast to the previous methodology we employed of using the minimum $^{238}\text{U}/^{232}\text{Th}$ as the local detrital $^{238}\text{U}/^{232}\text{Th}$, which was 0.22 at V30-99, 0.36 at U1313, and 0.42 at CDH19. The updated detrital $^{238}\text{U}/^{232}\text{Th}$ values do not alter our results in any way that would change the conclusions of this study (Fig. R4).

Figure R3. $^{234}\text{U}/^{238}\text{U}$ and $^{238}\text{U}/^{232}\text{Th}$ in the three cores of our study, V30-99 (A, B), U1313 (C, D), and CDH19 (E, F). Solid data points are those included in the detrital $^{238}\text{U}/^{232}\text{Th}$ calculation. Hollow data points are excluded. The horizontal dashed gray lines in (A), (C), and (E) are the $^{234}\text{U}/^{238}\text{U}$ threshold used to detect authigenic U. The data point in circle in (B) is excluded in the detrital $^{238}\text{U}/^{232}\text{Th}$ calculation because its $^{238}\text{U}/^{232}\text{Th}$ value is an outlier. Data points in circle in (C) have their associated $^{238}\text{U}/^{232}\text{Th}$ values excluded in the detrital $^{238}\text{U}/^{232}\text{Th}$ calculation because they are from a batch of samples with abnormally low ^{234}U . The horizontal dashed colored lines in (B), (D), and (F) are the resulting detrital $^{238}\text{U}/^{232}\text{Th}$.

Figure R4. Fig. 2 using the minimum $^{238}\text{U}/^{232}\text{Th}$ as the local detrital $^{238}\text{U}/^{232}\text{Th}$ (left) and using $^{234}\text{U}/^{238}\text{U}$ to detect authigenic U and calculate the local detrital $^{238}\text{U}/^{232}\text{Th}$ (right; as shown in the revised manuscript).

Reviewer #2 (Remarks to the Author):

This is the revised version of a study that I reviewed previously and that investigates the signature of sedimentary $^{231}\text{Pa}/^{230}\text{Th}$ extracted from three cores in the deep North Atlantic during the Last Glacial Inception (119-113 ka) in order to understand, how the AMOC responded to this climatic event that has, in contrast to glacial terminations, not received much attention despite that the next glacial inception will occur in the not so distant future.

The authors have added some new data for comparison to their data sets and despite that the number of data points capturing the LGI Pa/Th maximum is small, the possibility is high that there indeed was an AMOC instability during the LGI. I am still not 100% convinced that the data obtained from three cores undoubtedly reflect a short AMOC weakening of up to 1800 years but at the same time I cannot disprove it. I am therefore willing to consider the possibility that it occurred. Given the importance of such a finding I think that the community should now decide itself if this weakening may have occurred based on the presented data and at best produce further well resolved $^{231}\text{Pa}/^{230}\text{Th}$ records for this period of time to support or disprove the conclusions presented in this manuscript. In the end the $^{231}\text{Pa}/^{230}\text{Th}$ proxy is not supposed to show the same signal in each and every core but it is rather the basin-integrated ^{231}Pa deficit (low $^{231}\text{Pa}/^{230}\text{Th}$) that documents a strong AMOC whereas a lack or a decrease of the basin-integrated deficit will document an AMOC weakening (see original publication on the use of $^{231}\text{Pa}/^{230}\text{Th}$ as a paleo circulation tracer in the Atlantic Ocean by Yu et al. 1996). As such, further data will improve or disprove the reliability of the presented conclusions.

In summary, based on the above considerations I am now supporting publication of the manuscript.

Response #3 – We thank the reviewer for the positive assessment of our revised manuscript.

Minor suggestion for a change of the text in Line 161: This particle flux increase could....

Response #4 – Done.

Response to comments from reviewers

Key

- Reviewers' comments
- Authors' response

Reviewer #1 (Remarks to the Author):

After the previous review of the revised manuscript aimed at reconstructing the strength of AMO during Last Glacial Maximum from sedimentary Pa/Th records, the authors provided additional assessment of the particle type and rewording to address the influence of the particle flux on the interpretation of Pa/Th records. The authors also conducted reanalysis of the correction to obtain the scavenged fraction of the Pa/Th sediments, as suggested in the previous review, and updated the results in this revised version of the manuscript. Both of the work very well addressed the points raised in the previous review. I am happy to recommend the publication of this manuscript.

Response #1 – We thank the reviewer for the positive assessment of our work.

Minor:

Figure S6, and Figure S7: units for particle fluxes are incorrect, missing subscript (Fig. S6), and having typos (Fig. S7).

Response #2 – We thank the reviewer for catching the typos. We have corrected the particle flux units in both figures (see below).

Supplementary Fig. 6. Pa/Th and particle flux results. (A, C, E) Pa/Th data during the LGI. (B, D, F) Particle flux during the same period.

Supplementary Fig. 7. Scatter plots of particle flux and Pa/Th in the three cores of our study, V30-99 (A), U1313 (B), and CDH19 (C). The inset equations are the least square linear regression fit and the associated uncertainty. The pink symbols in each plot are the data points during the LGI interval.